# The impact of providing hiding spaces to farmed animals: A scoping review

**Hannah B. Spitzer**[1], **Rebecca K. Meagher**[2], **Kathryn L. Proudfoot**[1]*

**1** Health Management, Atlantic Veterinary College, University of Prince Edward Island, Charlottetown, Canada, **2** Department of Animal Science and Aquaculture, Dalhousie University, Truro, Canada

* kproudfoot@upei.ca

**Data Availability Statement:** All relevant data are within the paper and its Supporting Information files.

**Funding:** HS was provided a stipend provided by the University of Prince Edward Island's Sir James

## Abstract

Many wild animals perform hiding behaviours for a variety of reasons, such as evading predators or other conspecifics. Unlike their wild counterparts, farmed animals often live in relatively barren environments without the opportunity to hide. Researchers have begun to study the impact of access to hiding spaces ("hides") in farmed animals, including possible effects on animal welfare. The aims of this scoping review were to: 1) identify the farmed species that have been most used in research investigating the provision of hides, 2) describe the context in which hides have been provided to farmed animals, and 3) describe the impact (positive, negative or neutral/inconclusive) that hides have on animals, including indicators of animal welfare. Three online databases (CAB Abstracts, Web of Science, and PubMed) were used to search for a target population of farmed animals with access to hiding spaces. From this search, 4,631 citations were screened and 151 were included in the review. Fourteen animal types were represented, most commonly chickens (48% of papers), cattle (9%), foxes (8%), and fish (7%). Relatively few papers were found on other species including deer, quail, ducks, lobsters, turkeys, and goats. Hides were used in four contexts: at parturition or oviposition (56%), for general enrichment (43%), for neonatal animals (4%), or for sick or injured animals (1%). A total of 218 outcomes relevant to our objectives were found including 7 categories: hide use, motivation, and/or preference (47% of outcomes), behavioural indicators of affective state (17%), health, injuries, and/or production (16%), agonistic behaviour (8%), abnormal repetitive behaviours (6%), physiological indicators of stress (5%), and affiliative behaviours (1%). Hiding places resulted in 162 positive (74%), 14 negative (6%), and 42 neutral/inconclusive (19%) outcomes. Hides had a generally positive impact on the animals included in this review; more research is encouraged for under-represented species.

## Introduction

Over the last century, farm animal industries have shifted from traditional, extensive ways of raising animals to predominantly indoor housing systems, often using barren pens or cages [1, 2]. This shift, known as intensification, allowed for increased efficiency of animal production;

Dunn Animal Welfare Centre, https://awc.upei.ca/ KP is Director of the Sir James Dunn Animal Welfare Centre and played a role in the study design, data collection and analysis, decision to publish and preparation of the manuscript.

**Competing interests:** The authors have declared that no competing interests exist.

however, intensification also sparked growing concerns about the welfare of farmed animals [1]. Concerns for the welfare of animals generally fall into three spheres: animals' health and biological functioning, affective states, and ability to live in a natural way [3]. Animal producers and veterinarians tend to be most concerned about animals' health and production as indicators of animals with good welfare [4–6]. On the other hand, citizens are generally concerned about animals' affective states (feelings or emotions, [3]) and ability to live naturally [4, 6]. Indeed, naturalness is an aspect of welfare that is often overlooked in many animal production systems, as farmed animals have limited ability perform natural behaviours [7].

A common natural behaviour that is seen in many wild animals, but is often not available for farmed animals, is the ability to hide. In the natural environment, animals perform hiding behaviors for a variety of reasons and under different contexts. For example, free-roaming ungulates have been demonstrated to separate from the herd before parturition to find natural cover like trees or tall grass, including white-tailed deer [8], elk [9], Alaskan moose [10], wild and feral goats and sheep [11, 12], and semi-wild Maremma cattle [13]. Additionally, some ("hider" species) will leave their young hidden for several days after birth [14]. These peri-parturient hiding behaviours are thought to be both an anti-predatory strategy, as well as aiding in the dam and infant forming a bond before introduction to the herd [15]. Periparturient seclusion and neonatal hiding has also been documented in other wild counterparts to domesticated farm animals. For example, European rabbit does [16], mink [17] and red fox [18] dams hide their offspring to prevent predation and infanticide. Similarly, wild fowl, such as red jungle fowl [19], wild turkeys [20], and king eider ducks [21] will also find a covered laying site for to avoid nest predation.

Wild animals will hide in dens, burrows, or other naturally secluded areas at times other than parturition. For example, predators such as wild red foxes [22] and American mink [23] reside in dens, which often take the form of natural hollows in the earth or repurposed rabbit dens. These dens are typically situated under trees or other areas on natural cover as to act as shelter and protection from higher predators [22, 24], and the animals often move frequently between several different dens [23]. Similarly, prey species such as rabbits live in dens much like mink and foxes, although the rabbit typically builds the den themselves [25]. Use of the natural environment for sheltering is also an important antipredator behaviour for wild fish [26] and crustaceans [27], particularly for juveniles. Likewise, visual cover in the form of trees, bushes, and tall grass is also a major component of the natural environment for wild fowl, which birds use to evade predators or aggressive conspecifics, for example in the Red Jungle Fowl [28] and the North Island blue duck [29]. Although it is less reported in the literature, some wild animals as well as laboratory rodents have also been described to seek hiding places or seclusion for other reasons, including during sickness and injury [30–32].

Giving farmed animals the opportunity to perform natural behaviours, such as hiding, is an important component of animal welfare for many stakeholders [4, 6]. Moreover, allowing the performance of some natural behaviors may also improve other aspects of welfare, such as affective states and biological functioning [36]. Because of these potential benefits, there has been increasing research on allowing farmed animals to perform natural hiding behaviours, despite few conventional management systems that currently allow for this behaviour. Thus, the overall goal of this scoping review was to describe the present literature exploring the provision of hiding spaces ("hides") to farmed animals, which we defined as a space to hide from sight by placing themselves or their young away from a group, pen mate, or an open area. The aims of this scoping review were to: 1) identify the farmed species that have been most used in research investigating the provision of hiding spaces, 2) describe the context in which hides have been provided to farmed animals, and 3) describe the impact (positive, negative, or neutral/inconclusive) that hides have on animals, including indicators of animal welfare.

## Materials and methods

### Protocol registration

No review protocol was registered for this paper.

### Eligibility criteria

**Populations and interventions.**   To be included in the review, the paper must have met the following criteria: 1) the population of animals were those commonly farmed in North America (Table 1), 2) the intervention of the study included the provision of an artificial or natural hiding place or comparisons between hiding space design, placement, or timing of placement, or, the study was observational but one of the main outcomes was a measurement of hiding behaviour, and 4) the study must have had at least one outcome related to animal behavior and/or welfare. We considered a hiding place to be a space to hide from sight by placing themselves or their young away from a group, pen mate, or an open area (e.g., shelters, nesting boxes, natural cover, visual barriers, or spaces to isolate from other animals). Studies that provided animals with shelters with the primary goal of assessing animals' sheltering from harsh weather were excluded. If a hide was included in the study but hiding behaviour was not included as an outcome, the study was excluded. Studies were also excluded if they reported nest building behaviours or interactions with nesting materials but hiding behaviour was not included as an outcome.

**Comparators and outcomes.**   Comparisons were made between species and stage of life (e.g., parturition, neonatal or other life stages). Outcomes investigated included any behavioral measurements well as other measurements that were related to animal welfare, including hide use, hide preference, indicators of physiological stress responses, health, and abnormal behaviour.

**Limitations.**   Only academic journal articles written in English were considered for inclusion. There were no limitations on year of publication or country of publication.

**Table 1.  Algorithm for database search.**

| Operator | Concept Search String |
|---|---|
| | (cattle or cow or cows or calf or calves or bull or bulls or heifer or heifers or steer or steers or deer or doe or does or hind or hinds or stag or stags or buck or bucks or fawn or fawns or kid or kids or pig or pigs or sow or sows or hog or hogs or boar or boars or piglet or piglets or bison or elk or sheep or ewe or ewes or ram or rams or lamb or lambs or goat or goats or nanny or nannies or billy or billies or alpaca or alpacas or llama or llamas or horse or horses or mare or mares or stallion or stallions or gelding or geldings or foal or foals or donkey or donkeys or jenny or jennys or jack or jacks or chicken or chickens or hen or hens or rooster or roosters or cockarel or cockarels or pullet or pullets or chick or chicks or duck or ducks or drake or drakes or duckling or ducklings or turkey or turkeys or tom or toms or gobbler or gobblers or jake or jakes or poult or poults or quail or quails or mink or minks or kit or kits or pup or pups or fox or foxes or vixen or vixens or fish or rabbits) |
| AND | (retreat or hide or hides or hiding or wall or nest or nesting or seclude or secluding or seclusion or shelter or sheltering or cover or covered or brood or refuge or blind or blinds or box or protect or protected or enclosure or enclosed or hidden or "birth-site" or "nest-site" or "nesting site" or "maternity site" or "maternity pen" or "farrowing site" or "calving site") |
| AND | (welfare or behaviour or behavior or health or housing or environment or stress or calving or parturition or neonatal or sick or ill* or natural or farm or management or habitat or landscape or motivat* or prefer* or lay* or prey* or pain* or enrich*) |
| AND | (welfare or housing or enrich or enrichment or wellbeing or "well-being" or stress) |
| AND | (behaviour or behavior) |
| NOT | ("box stall" or "box-stall" or microb* or bacter* or infect* or parasit* or "animal shelter" or oxidat* or mouse or mice or rat or rats) |

## Search

With the assistance of a librarian experienced in scoping reviews and veterinary literature, searches were designed in three databases: CAB Abstracts (via EBSCOhost), PubMed, and Web of Science. See Table 1 for full details of the search terms and construction. Concepts were searched in title and abstract fields, and a single search for all terms was run in each database. Searches were conducted on June 4, 2021.

Three databases were searched (CAB Abstracts, PubMed, Web of Science) with the aim of searching for experimental papers with a population of farmed animals and an intervention of hiding place. Some search terms were repeated; this was done with the guidance of a librarian to reduce irrelevant search results.

## Selection

Fig 1 outlines the screening procedure for paper inclusion using a modified PRISMA 2020 flow diagram [33]. From the search results, citations were uploaded to the Rayyan QRCI web application [34] to be manually screened for inclusion. Inclusion and exclusion criteria were flexible and were discussed between authors to reach consensus. One author (HBS) first screened papers based on title and abstract, and excluded papers based on incorrect population (animals farmed in North America) or incorrect intervention (provision of a hiding place; papers that only assessed nesting-building behaviours were excluded), incorrect publication type (reviews, books, and conference proceedings were not included), weather (studies that provided animals with shelters with the primary goal of assessing animals' sheltering from harsh weather were excluded), and incorrect language (non-English languages). After the papers were screened by title and abstract, the same author (HBS) reviewed the remaining papers by full text using the same inclusion/exclusion criteria. After full text screening, the reference lists of the remaining papers were then hand searched for any additional relevant papers. Although the population of interest was animals commonly farmed in North America, the study did not have to be conducted in North America.

## Coding and appraisal

The 151 papers included in the final review were coded to facilitate analysis of the aims of this paper. All papers were reviewed by full text and were coded in each of the following categories: Title, author, year of publication, animal category (ungulate, poultry, fur carnivore, small herbivore, fish, crustacean), animal type (cattle, deer, pigs, sheep, goats, chickens, quail, ducks, turkeys, mink, foxes, fish, lobsters, and rabbits), context of hide provision (e.g., timepoint in the animals' life when the hide was provided), and study outcomes (described using notes). We chose to use these specific "animal types" due to their relevance to farm animal industries. We also opted to further group them into "animal categories" based on biological and behavioural similarities; for example, we used the broader category of ungulate to represent cattle, deer, pigs, sheep, and goats because these animals have similar natural histories. However, rabbits and lobsters were the only animal types found in our search that fit in the small herbivore and crustacean categories.

Study outcomes were first described by one author (HBS) using notes for each paper. These outcomes were then subjectively categorized by topics that emerged (e.g., specific behavioural or physiological data that were collected). The outcomes were then subjectively classified as representing the hide having "positive", "negative" or "neutral/inconclusive" impacts on the study animals. Outcomes that matched at least one of the following criteria were considered to reflect a positive impact: 1) animals used the hides, were motivated to access the hides, or preferred hide design features that increased the level of seclusion compared to a "standard" hide (for example, a nest box with a curtain compared to a standard nest box without a curtain); 2)

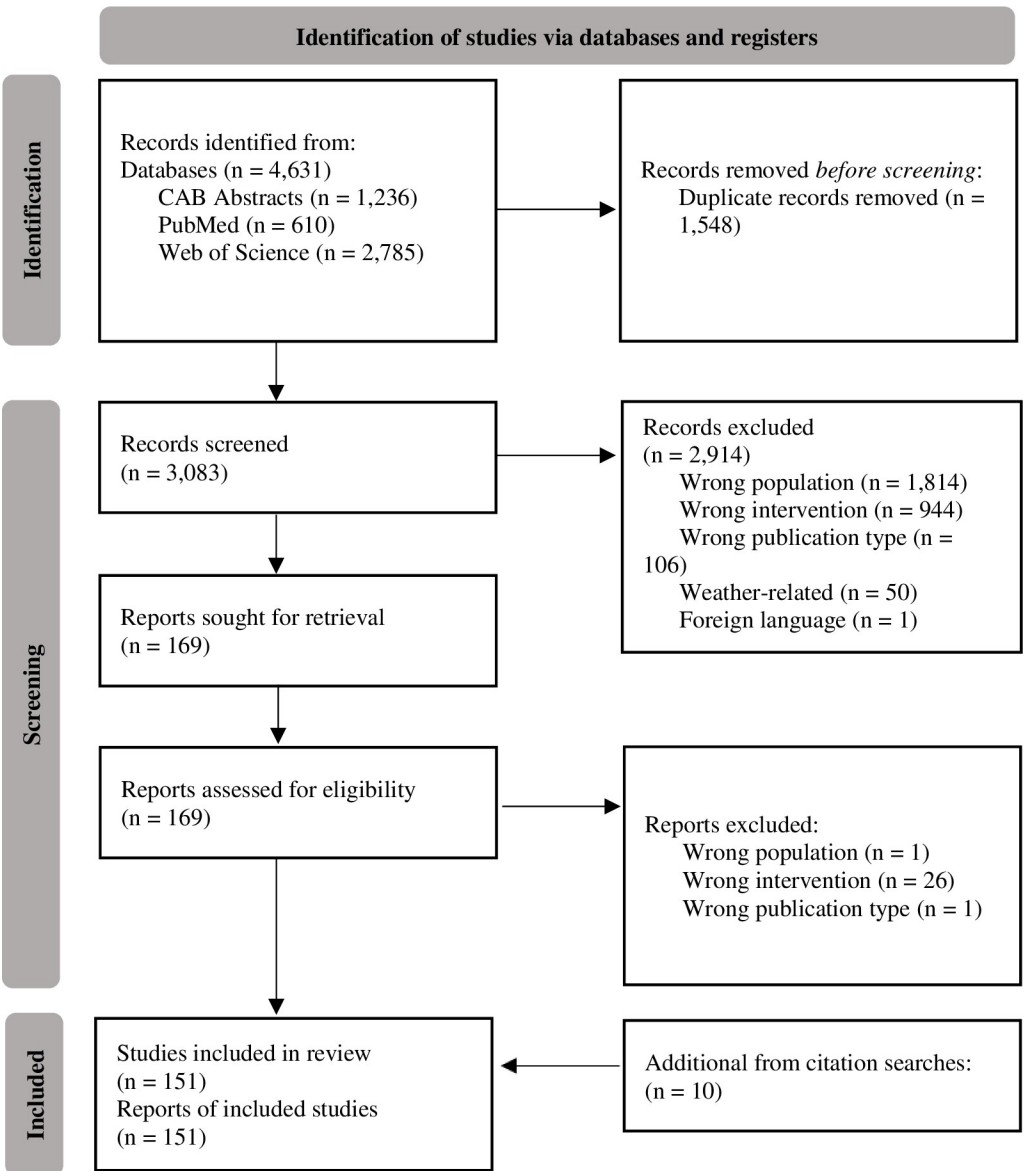

**Fig 1. Screening procedure and exclusion process for scoping review.** The aim was to search for experimental papers with a population of farmed animals and an intervention of hiding place.

animals increased performance of behaviours that typically indicate positive welfare, such as affiliative behaviours or other behavioural indicators of affective state (e.g. preening); 3) animals reduced performance of behaviours that indicate negative welfare, such as agonistic behaviours, abnormal repetitive behaviours or other behavioural indicators of negative affective state (such as behavioural indicators of fear); or 4) there was a decrease in injuries or physiological indicators of stress, or an increase in health or productivity.

Study outcomes that matched at least one of the following criteria were considered to reflect a negative impact of hides on the animals: 1) animals decreased performance of behaviours that indicate positive welfare, such as affiliative behaviours or other behavioural indicators of positive affective state (such as preening); 2) animals increased performance of behaviours that indicate negative welfare, such as agonistic behaviours, abnormal repetitive behaviours, or other behavioural

indicators of negative affective state (such as behavioural indicators of fear); or 3) there was an increase in injuries or physiological indicators of stress, or a decrease in health or productivity.

Some outcomes did not indicate that animals benefitted from the hides but also did not suggest that the animals were negatively impacted by the hides; these were classified as neutral or inconclusive and included the following: 1) animals did not use the hides, did not demonstrate motivation to access the hides, or did not show preference for hide design features that increased seclusion; 3) rate of injuries, agonistic behaviours, or abnormal repetitive behaviours were unaffected by hide provision; 2) conflicting behaviours of similar type were reported (for example, one type of abnormal repetitive behaviour was decreased when hides were provided but another abnormal repetitive behaviour was increased); or 3) conflicting physiological results were reported (for example, one type of injury was decreased when hides were provided but another type of injury was increased).

## Results

### Search results and exclusions

A total of 4,631 papers were retrieved (CAB: 1,236; WOS: 2,785; PubMed: 610). Duplicates (n = 1,548) were immediately detected and eliminated (Fig 1). A total of 3,083 papers were reviewed by title and abstract; 2,914 were excluded. Of these, 1,814 papers were initially excluded because the study population was not farmed animals. A further 944 were excluded because the intervention was unrelated to hiding behaviour and 106 were excluded because the publication was not a peer-reviewed journal article. Fifty papers were excluded because the primary aim was related to sheltering from harsh weather, and 1 foreign language paper was excluded. A total of 169 papers were reviewed by full text. A further 28 papers were excluded: 1 for wrong population, 26 for wrong intervention, and 1 for wrong publication type. The reference lists of the included papers were hand-searched for missing papers relevant to this review; 6 papers were added through initial hand searching. A reviewer suggested the inclusion of a further 2 papers; these were included, and the reference lists of these papers were hand-searched for relevant papers, leading to inclusion of a further 2 papers. After this process, a total of 151 papers were included in the final review. Publication years ranged from 1976–2021.

### Aim 1: Animal types represented in the literature

Fourteen types of farmed animals were represented in the 151 papers (Fig 2). The most represented animals were chickens (48% of papers), cattle (9%), foxes (8%), and fish (all species pooled, 7% of the papers). Many animal types were also be split into subtypes, which reflected differences in hiding behaviour. For example, the group of chicken papers were composed of laying hens (84% of the chicken papers) as well as broiler chickens (15% of the chicken papers). The cattle papers were nearly all focused on dairy cattle (92% of the cattle papers), while only two papers (15% of the cattle papers) investigated beef cattle (one of which focused on both dairy and beef cattle, while the other investigated beef calves).

One paper represented both turkeys and ducks, so this paper was categorized as both a 'duck' paper and a 'turkey' paper, resulting in 152 total papers in Fig 2. There were 6 animal types included in the initial literature search for which no results were found: bison, elk, alpacas, llamas, horses, and donkeys.

### Aim 2: Context of hide provision

Four categories of hide provision were found: 1) at parturition or oviposition (56% of the papers), 2) for general enrichment (43% of the papers), 3) for neonatal animals (4% of the

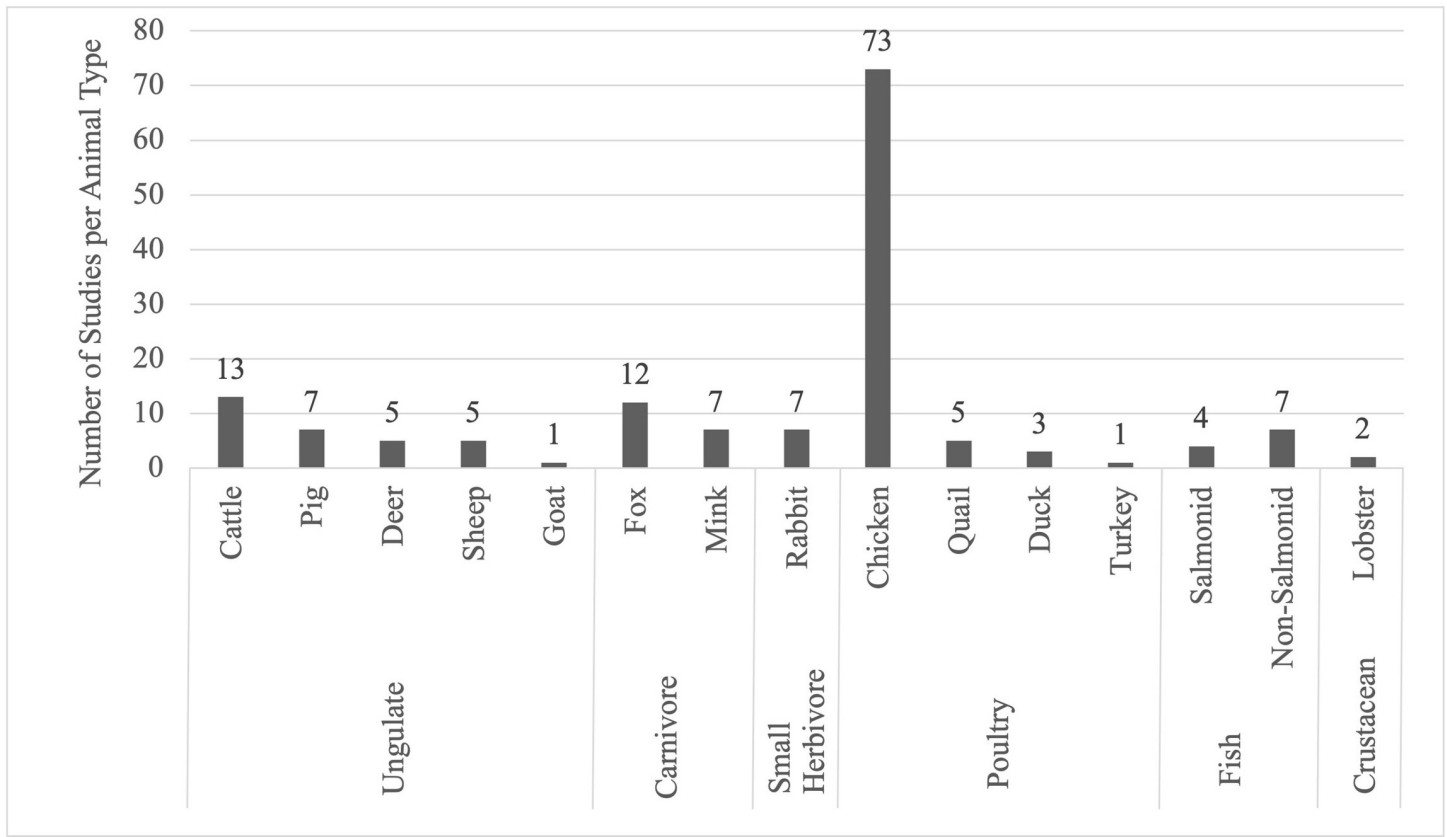

**Fig 2. The number of studies per category and type of animal.** From the 151 papers included in this scoping review, there were 6 categories (e.g., "ungulate") and 15 types (e.g., "cattle") of animals represented. This figure shows the number of studies for each of the animal categories and types. One paper represented both turkeys and ducks, so this paper was categorized as both a 'duck' paper and a 'turkey' paper, resulting in 152 total papers in the graph.

papers), or 4) for sick or injured animals (1% of the papers). Some papers (n = 6) represented two categories of hide provision; these papers were included in both categories. The context of the hide use seemed to be dependent on the animal type (Fig 3). Most papers that focused on parturition and oviposition included ungulates and poultry, all papers that focused on neonatal animals and sickness or injury were focused on ungulates, and there were a variety of species included in papers using hides as general enrichment.

## Aim 3: Effect of hide provision on the animals

A total of 218 study outcomes were coded from the 157 studies, as many studies had more than one reported outcome. A total of 7 categories emerged from the study outcomes, including: 1) 'hide use, motivation, and/or preference', which included outcomes related to animals' use of provided hides, their motivation to access to hides, or preference of certain hide features, 2) 'affiliative behaviour', such as allogrooming, 3) 'agonistic behaviour', including aggression, threatening behaviour, or retreating, 4) 'abnormal repetitive behaviour', such as repetitive pacing or feather pecking, 5) 'health, injuries, and/or production', such as rates of illness, injuries, or growth, 6) 'physiological indicators of stress' such as corticosterone levels, and 7) 'behavioural indicators of affective state, such as preening or settled pre-laying or pre-parturient behaviour.

Table 2 shows the distribution of study outcomes into positive, negative, and neutral/inconclusive categories, along with the number of outcomes that were found for each context of

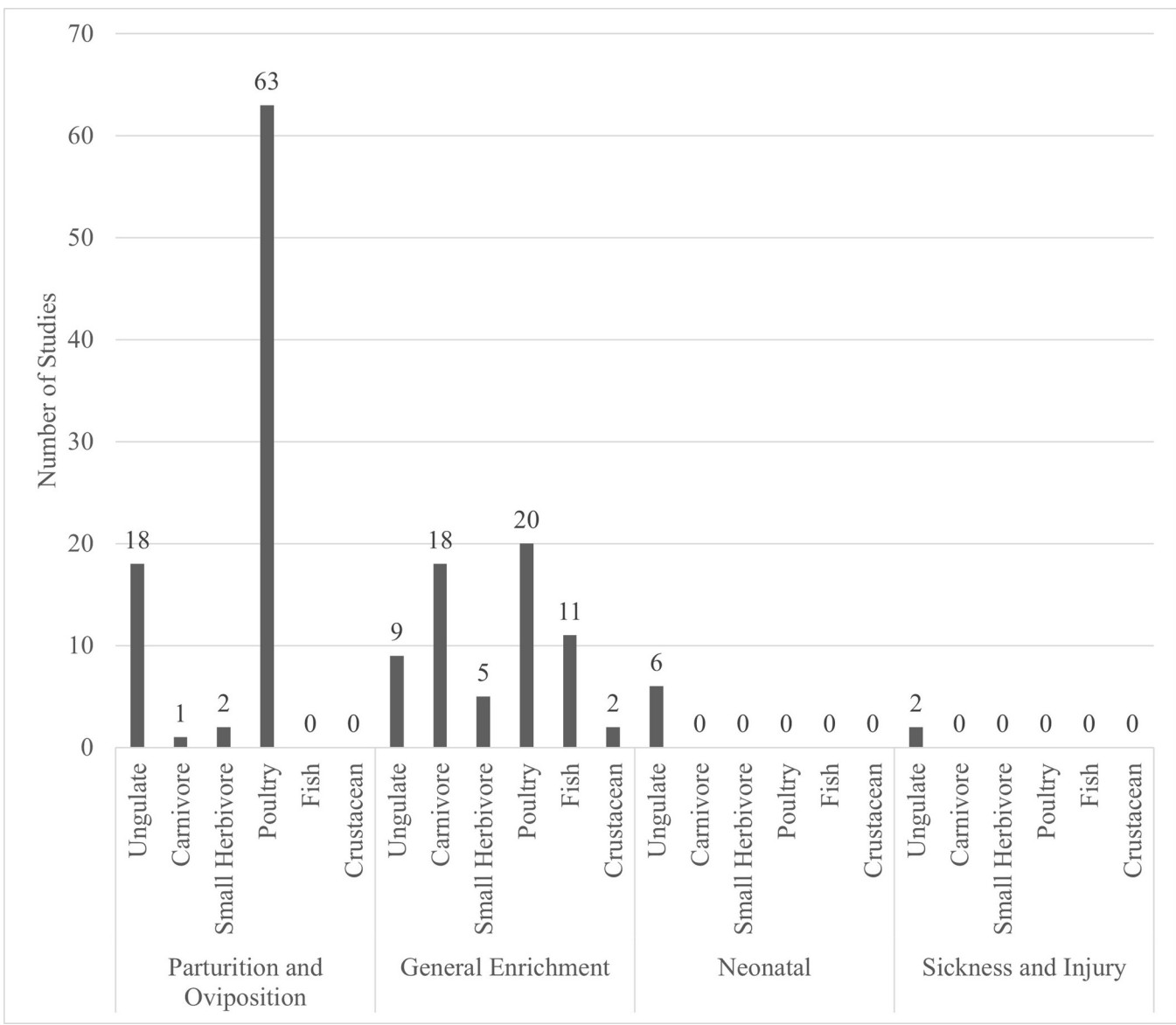

**Fig 3. The number of studies per category of animal and context of hide provision.** From the 151 papers included in this scoping review, there were 6 categories of animals (e.g., "ungulate") and 4 contexts of hide provision (e.g., "parturition and oviposition") represented. This figure shows the number of studies for each of the animal categories and context of hide provision. Three papers represented two contexts of hide provision; these papers were included in both categories for a total of 157 papers represented in the figure.

hide provision (aim 2) per animal category. Overall, there were 162 positive outcomes, 14 negative outcomes, and 42 neutral/inconclusive outcomes. Of 151 papers, 103 papers reported outcome related to hide use, motivation, and/or preference, 2 papers reported outcomes related to affiliative behaviour, 17 papers reported outcomes related to agonistic behaviour, 14 papers reported outcomes related to abnormal repetitive behaviour, 35 papers reported outcomes related to health, injuries, and/or production, 11 papers reported outcomes related to physiological indicators of stress, and 37 papers reported outcomes related to behavioural indicators of affective state.

The number of papers reporting that hide provision resulted in a positive, negative, or neutral/inconclusive impact on the animals is listed for each category. A '+' indicates that hide

**Table 2. Categorization of 151 papers based on how provision of a hiding space impacted the animals.** The number of outcomes per context of hide provision is listed under each animal category. Missing categories indicates that there were no outcomes found in that context for that species; a blank cell indicates that no outcomes were found for that animal category and context of hide provision.

| | Hide Use, Motivation, and/or Preference | | | Affiliative Behaviour | | | Agonistic Behaviour | | | Abnormal Repetitive Behaviour | | | Health, Injuries, and/or Production | | | Physiological Indicators of Stress | | | Behavioural Indicators of Affective State | | |
|---|---|---|---|---|---|---|---|---|---|---|---|---|---|---|---|---|---|---|---|---|---|
| | + | - | +/- | + | - | +/- | + | - | +/- | + | - | +/- | + | - | +/- | + | - | +/- | + | - | +/- |
| **Ungulates:** *Total* | 17 | 0 | 5 | 1 | 0 | 0 | 5 | 2 | 0 | 2 | 0 | 0 | 5 | 0 | 1 | 1 | 0 | 0 | 2 | 1 | 0 |
| *Parturition* | 10 | | 5 | | | | | | | 1 | | | 2 | | | 1 | | | 2 | | |
| *General Enrichment* | 1 | | | 1 | | | 5 | 2 | | 1 | | | 3 | | 1 | | | | | 1 | |
| *Neonatal* | 4 | | | | | | | | | | | | | | | | | | | | |
| *Sickness or Injury* | 2 | | | | | | | | | | | | | | | | | | | | |
| **Poultry:** *Total* | 51 | 0 | 12 | 0 | 0 | 0 | 2 | 1 | 1 | 3 | 0 | 0 | 11 | 0 | 1 | 5 | 0 | 2 | 17 | 0 | 6 |
| *Oviposition* | 36 | | 8 | | | | 2 | 1 | 1 | 3 | | | 10 | | 1 | 5 | | 2 | 16 | | 6 |
| *General Enrichment* | 15 | | 4 | | | | | | | | | | 1 | | | | | | 1 | | |
| **Fur Carnivores:** *Total* | 6 | 0 | 3 | 0 | 0 | 0 | 1 | 0 | 0 | 6 | 0 | 0 | 5 | 2 | 0 | 1 | 0 | 0 | 3 | 2 | 2 |
| *Parturition* | | | | | | | | | | | | | 1 | | | | | | 1 | | |
| *General Enrichment* | 6 | | 3 | | | | 1 | | | 6 | | | 4 | 2 | | 1 | | | 2 | 2 | 2 |
| **Small Herbivores:** *Total* | 0 | 0 | 2 | 0 | 0 | 1 | 0 | 0 | 2 | 1 | 0 | 0 | 2 | 1 | 2 | 0 | 0 | 0 | 2 | 0 | 0 |
| *Parturition* | | | | | | | | | | | | | 1 | | 1 | | | | 1 | | |
| *General Enrichment* | | | 2 | | | 1 | | | 2 | 1 | | | 1 | 1 | 1 | | | | 1 | | |
| **Fish:** *Total* | 5 | 0 | 0 | 0 | 0 | 0 | 1 | 2 | 0 | 2 | 0 | 0 | 3 | 1 | 2 | 1 | 1 | 0 | 0 | 1 | 0 |
| *General Enrichment* | 5 | | | | | | 1 | 2 | | 2 | | | 3 | 1 | 2 | 1 | 1 | | | 1 | |
| **Crustaceans:** *Total* | 2 | 0 | 0 | 0 | 0 | 0 | 0 | 0 | 0 | 0 | 0 | 0 | 0 | 0 | 0 | 0 | 0 | 0 | 0 | 0 | 0 |
| *General Enrichment* | 2 | | | | | | | | | | | | | | | | | | | | |
| **Overall Total** | 81 | 0 | 22 | 1 | 0 | 1 | 9 | 5 | 3 | 14 | 0 | 0 | 26 | 4 | 6 | 8 | 1 | 2 | 24 | 4 | 8 |

provision had a positive impact on the animals, a '-' indicates that hide provision had a negative impact on the animals, and a '+/-' indicates that hide provision did not impact animals or that the results were otherwise inconclusive.

## Discussion

The overall goal of this scoping review was to describe the existing literature exploring the provision of hiding places to farmed animals. We identified that chickens were the most used animal in research exploring hide use, and that hides were used mainly for parturition/oviposition and for general enrichment. Further, we found that hides resulted in primarily positive outcomes related to animal behaviour and/or welfare.

### Aim 1: Animal types represented in the literature

A wide variety of farmed species were included in this review, but much of the focus was on species most used for food production and for domestic animal welfare research. For example, about half (73/151) of the papers included in this review investigated the hiding behaviour of chickens, with a majority of these focused on laying hens. Chickens are not only one of the most widely used food animals globally but have also a long history of being the focus of animal welfare research (for example on laying hens, see [35]). A moderate number of studies (6 to 13 studies per species listed) were performed on other common farm animals such as cattle, foxes, fish, pigs, sheep, and mink.

Relatively little research (1 to 5 studies per species listed) has been conducted on other farm animal species including deer, quail, ducks, lobsters, turkeys, and goats. The lack of research

on this topic for these animal types is a gap in the existing literature. Despite the lack of research in farm settings, these animals are still likely to have an instinct to hide. For example, ducks are known to have a natural drive to find a secluded nesting area to lay their eggs [21]. Wild lobsters also make great use of hiding places throughout their lives, especially in the juvenile stage when lobsters will near-continuously remain in or near a shelter to avoid predation [27]. Wild goats are known to seclude from their herd during the periparturient period, and neonatal kids will remain hidden for the first several days after birth [36].

Six animal types were included in the initial literature search as farmed species in North America, but no papers were found (bison, elk, alpacas, llamas, horses, and donkeys). Literature describing the natural behaviour of bison and elk indicates that they have many of the same natural hiding instincts as the other animals included in this review, particularly during the periparturient and neonatal periods [37, 38]. Despite their prevalence as farmed animals, little information is available on the natural hiding behaviour of horses, donkeys, alpacas, and llamas; more investigation is warranted. We encourage more research on providing hiding spaces to these animals, especially for those where there is evidence to suggest the species would use a hiding space in a more natural setting.

## Aim 2: Context of hide provision

**Parturition and oviposition.** More than half (84/151) of the papers focused on the time of parturition and oviposition; most of these papers (63/84) explored the use of nest boxes at oviposition by poultry. In these papers, common methodologies included investigation on whether hens make use of nest boxes, the factors that influence nest box use, and the impact that nest box provision has on the hens. For example, one early study [39] explored all three of these factors in caged laying hens by providing hens with various configurations of nest box and describing the chickens' use of the nest boxes along with their behaviour around oviposition.

Several papers (18/84) investigated the use of hiding spaces around parturition in ungulates. Most of these (11/18) were focused on dairy cattle, typically describing cows' use of spaces to seclude from pen-mates at parturition. Spaces to seclude before, during, or after calving were provided in several different ways; for example, creating enclosed box stalls with wide or narrow entrance gaps [40] or placing a hiding wall [41] in group maternity pens, or providing areas of natural cover [42]. The remaining ungulate papers described the periparturient seclusion behaviour of pigs [43] and sheep provided with artificial [44–46] and natural hiding spaces [47], as well as farmed deer provided with both artificial covered areas [48] as well as areas of natural cover [49]. A further 3 papers explored the periparturient hide use of farmed foxes [50] and rabbits [51, 52] when provided with nest boxes that allowed for relatively more seclusion compared to a traditional nest box.

Allowing periparturient animals the space to seclude common in the presented literature. However, more research could be beneficial, particularly in animals that are not well-represented in the current literature. For example, few papers explored the provision of nest boxes to non-chicken poultry, particularly ducks and turkeys, despite evidence that these animals seek seclusion around oviposition in the natural environment [21, 53]. Likewise, non-cattle ungulates such as goats and pigs also demonstrate the natural urge to seclude around parturition [36, 54, 55]; however, no papers were found on this topic for goats and only two papers were found for pigs. Many papers have studied the expression of natural nest-building behaviours in farmed sows (for a review, see [54]), but more research investigating sows' or piglets' desire to physically hide or seclude could be beneficial. Four papers were found on the seclusion of ewes around lambing; however, all were published in or before 1985, thus, more

research into current farming settings is encouraged. Mink, foxes, and rabbits are known to seclude during the periparturient period in the natural environment [18, 56, 57]; however, relatively few papers were found exploring this behaviour in the farmed counterparts of these animals.

**General enrichment.** The second most common context for hides was their use for general enrichment for juvenile and adult animals (65/147), with all animal types represented. Many (20/65) of these papers explored the use of hiding spaces for poultry and were often placed in an outdoor range as a method to promote greater ranging behaviour in the birds. This was done using both artificial structures, such as vertical and/or horizontal walls [58], as well as natural vegetative cover [59]. Providing hiding shelters, including hay bales [60] and vertical walls [61], to poultry in indoor pens was also done to investigate their use to promote birds' comfortable use of the space, particularly in the center of pens.

Fur carnivores, including farmed foxes and mink, were the next most represented animal type in this category of papers (18/65). Farmed foxes and mink are unique, as it is often considered standard for modern farming to provide these animals with hiding boxes in the cage [23]. Thus, the papers found for this review primarily focus on the impact of giving foxes and mink additional hiding places, often in combination with other enrichment items. These additional hiding places often took the form of an extra nest box [62] or even a mesh tunnel [63]. Some papers, particularly early papers, did investigate the benefits of providing these animals with a sole hiding place compared to a barren cage [64, 65]. Assessments of animals' motivation to access nest boxes (along with other enrichment items) were also performed [66], as were descriptions of how much the animals' interacted with hiding places compared to other enrichment items [66, 67] and how much the animals used the hiding places over time [68], which tended to be higher and remain high over time compared to other enrichment items.

Fish and crustaceans were also represented in this category (13/65). These papers described the animals' use of hiding places in otherwise barren tanks and how the animals were affected by the hiding places. Hides were typically provided to group-housed fish in the form of tunnels [69], artificial plants [70], real plants [71], or other overhead cover [72]. Hides were provided to singly housed lobsters in the form of a dish with sides [73], a box [73], or a tunnel [74]. The benefit of rearing fish and lobsters with hiding places on survival rates upon release into the wild was also explored [74, 75], as this may be a useful method to "train" these animals to seek safety from potential predators as they would in the natural environment.

Several papers (9/65) explored the provision of hiding places for enrichment to ungulates, including pigs, goats, sheep, and cattle. These papers primarily focused on the provision of hiding walls [76] or cubicles [77] to group-housed animals to decrease agonistic behaviour between animals. Several more papers (5/65) explored the provision of hiding places to small herbivores, which explored group-housed rabbits' use of hiding areas, including tubes [78], walls [79], and boxes [80], and the subsequent impact on the rabbits, particularly agonistic and abnormal behaviours. Overall, the provision of hiding places to animals as general enrichment has been well researched as a tactic to improve animal welfare across many types of animals, particularly in ungulates, fur carnivores, poultry, and fish.

**Neonatal, sickness, and injury.** Providing hiding spaces to neonatal and sick/injured animals was less common in the literature, with only 6 and 2 papers existing on each these topics, respectively. All 8 of these papers focused on ungulates. Four of the papers exploring hide use in neonatal animals focused on newborn, extensively housed deer calves, and described calves' use of both artificial [81] and natural [82] hiding spaces in the days immediately following birth. Two papers described how indoor-housed dairy cow-calf pairs utilized barriers in the pen as a place to seclude in the hours after birth [40, 83]. More research is encouraged to assess neonatal hiding behavior in farmed species, particularly ungulates that are considered to be

"hider" species, such as cattle and goats, where mothers will hide their young depending on their available resources [15]. Oftentimes "follower" species, such as sheep, are thought to perform fewer neonatal hiding behaviours as other "hider" species [84]. However, investigation into neonatal hiding is still warranted, as the hider-follower dynamic has been criticized as overly simplistic (see [15]) and free-ranging sheep also show similar seclusion seeking behaviours to cattle in the peri-parturient and neonatal stages [55]. Both papers exploring hiding during illness and injury were focused on dairy cattle, including cows' patterns of seclusion during times of periparturient sickness [85], and calves' use of hiding spaces after disbudding [86]. In many farm environments, animals may be isolated in hospital pens which may give them some level of seclusion; seclusion may be worth considering when designing hospital areas for sick and injured animals. We encourage further research on this topic, as separation from conspecifics is considered a "sickness behaviour" in laboratory animals [31], and may have valuable applications to animal production industries.

## Aim 3: Effect of hide provision on the animals

**Hide use, motivation, and/or preference.** Overall, it appears that most authors reported positive outcomes for animals in their studies using hides. Many of these papers focused on use, preference, or motivation for a hiding space. Hide use was a common outcome in many species, particularly around parturition and oviposition. For example, nearly all papers reported that laying chicken (both cage [39] and aviary [87] housed), quail [88], and duck [89] hens used provided nest boxes to lay their eggs. Likewise, extensively housed deer [49], extensively housed cattle [90], indoor-housed dairy cows [41], dairy cow-calf pairs [83], semi-extensively housed dairy cattle [42], indoor-housed ewes [45, 46], and outdoor-housed ewes [44] were reported to retreat to secluded hiding areas before, during, and after parturition. Indoor-housed dairy cows increased use of hiding spaces when experiencing illness during the post-parturient period.

Some inconclusive/neutral results for dairy cows, sows, and ewes were also reported, as some papers reported that most cows [91, 92], sows [48], and ewes [47] did not make use of provided hiding areas for parturition. It is possible that there are also other factors influencing the dams' choice of calving, farrowing, or lambing site; for example, one study found that cows appeared to choose calving sites that were close in proximity to the location of a previous calving [92]. Additionally, some poultry papers report that a minority of hens who do not use nest boxes ("floor layers") may find provided nest boxes to be unsatisfactory [93, 94]. However, others posit that these floor layers may simply prefer more open nest sites and are equally motivated to access these open nest sites compared to nest layers [95]. Research into how to design satisfactory facilities for all hens could help improve the welfare of the animals as well as improve productivity, as floor laying is a concern for aviary housing systems in which the eggs are collected via the nest boxes.

Many papers reported animals used hiding structures for times other than oviposition/parturition. For example, newborn, extensively housed deer calves were also reported to make great use of both natural [82] and artificial [96] hiding places to hide in the days following birth. Indoor-housed dairy calves also used a secluded hiding area in a group pen, and increased hiding behaviour when experiencing injuries after disbudding [86]. Group-housed pigs make extensive use of hiding partitions in group pens, both during feeding and resting [97]. Both farmed mink [63] and foxes [67] have been reported to make use of hiding places and use remained high over time [68]. However, some papers reported that farmed foxes did not use provided hiding spaces [98], and that the roof of the nest was more often used as a platform instead. For example, one paper reported that mink preferred to rest in a mesh, non-

opaque overhead tunnel that connected the main pen to the enrichment pen, rather than in the opaque hiding areas provided in the enrichment pen and in the home pen [63].

Only two outcomes related to hide use in rabbits were reported, both of which were inconclusive. One paper reported that rabbits did not use a provided hiding tube [78], and another reported that rabbits did use a provided shelter, but it was mostly used as a platform rather than as a hiding place [80]. More research into the provision of hiding places for rabbits is warranted to be able to provide structural enrichment that is meaningful for the animals. For example, wild rabbits dig their own burrows [25] whereas the studies found in this review provided rabbits with pre-made hiding places. The evidence that foxes, mink, and rabbits may prefer to rest on tall places rather than inside enclosed hiding areas also warrants further research into preferred hiding place design for these animals.

Papers involving fish and crustaceans reported 7 outcomes related to hide use, all of which were positive. Papers reported that many species of fish, including char [99], wrasse [100], bream [75], catfish [101], and cod [102], as well as lobsters [73, 74], used hiding places that were provided to them. Since the results demonstrated that the studied fish made use of hiding structures when provided, continued research on other fish species as well as on preferred hide design features is encouraged.

Juvenile and adult poultry also utilized hiding structures for times other than oviposition: Broiler chicks were reported to cluster around straw bales provided for shelter [60, 103], chickens clustered around both vertical and overhead hiding structures in an outdoor range [58, 104], and caged quail made use of provided hiding walls [105]. Despite this, some poultry papers reported inconclusive findings. For example, placing visual barriers around dust baths did not improve use of them [106], a few papers reported that hiding places in the range did not improve use of the range [107], and placing vertical hiding walls in the middle of a pen did not encourage birds to use the center of the pen [108]. Although most research suggests that poultry make extensive use of hiding structures for general enrichment, it remains unclear why some hiding structures are not used.

Motivation tests were also used to assess animals' desire to access hiding places. Chicken [95, 109] and duck [89] hens demonstrated motivation to access nesting boxes for oviposition by pushing through increasingly narrow gaps and weighted push doors to access the nest boxes. Foxes also demonstrated motivation to access hiding areas by pushing weighted doors [66, 110]. However, dairy cows did not demonstrate motivation to access secluded calving areas around parturition; the authors posited that the cows could not demonstrate the learned push-door task in combination with the desire to access the secluded area during calving [91].

Factors influencing animals' preference of various hide features were also explored, particularly in poultry. Laying hens most often preferred the most secluded nest available. This included top rack or corner nests in aviaries [111] and nests with enclosed sides or curtained entrances [112, 113]. Extensively housed deer calves also demonstrated a preference for natural hiding places over artificial hiding places when offered simultaneously, although the artificial hides were also used [81]. Studies have also investigated dairy cows' preference for narrow or wide hiding walls; the wider walls were preferred by both cows and cow-calf pairs [40]. This research suggests that animals will often prefer hiding areas with relatively greater degrees of seclusion.

**Affiliative and agonistic behaviours.** Some papers measured the impact of hide use on social behaviors, such as increasing affiliative behaviours and reducing agonistic behaviour. Two papers (1 positive and 1 neutral) reported outcomes related to affiliative behaviours: a hiding wall increased affiliative behaviours between group-housed calves [114], while a hiding structure decreased overall social contact between group-housed rabbits [115].

Regarding agonistic behaviour, 9 positive outcomes, 5 negative outcomes, and 3 neutral outcomes were reported. Hiding places were demonstrated to decrease agonistic behaviours between socially housed goats [116], pigs [97, 117], calves [114], quail [118], foxes [119], and fish [70]. However, several papers reported that providing group-housed animals with hiding places increased agonistic behaviours, this was reported to have resulted from animals competing for the hiding spaces [69, 100, 120] or because the hide formation subsequently restricted open space in the pen [76, 121]. Based on the existing literature on this topic, it appears that giving group-housed animals spaces to hide can improve the social interactions between the animals, but only if implemented in a way that does not cause strain on the space allowances of the living space.

**Abnormal repetitive behaviours.**   Fourteen outcomes related to abnormal repetitive behaviours were reported, all of which were positive, indicating that these behaviors were reduced when a hide was provided. For example, providing animals with hiding areas for general enrichment or nest boxes for oviposition decreased abnormal repetitive behaviours in pigs [121], laying hens [122], foxes [68], mink [123], rabbits [80], and fish [70]. It was commonly concluded that the reduction of abnormal repetitive behaviours was because hiding spaces helped reduce affective and physiological states that are presumed to be negative, such as perceived boredom, distress, or fear.

**Health, injuries, and/or production.**   Some papers found that the provision of the hide improved measurements of health, reduced injuries, or improved productivity. Of the papers that were found, 24 positive outcomes, 4 negative outcomes, and 6 neutral outcomes were reported. Many papers reported that animals, including pigs, hens, foxes, mink, and fish provided with hiding places had increased feed intake [75], improved growth [99, 123], higher egg production [124], improved mating success [125], fewer injuries (including injuries from the environment as well as conspecific-inflicted and self-inflicted mutilation) [77, 126, 127], and decreased mortality [50, 99]. Two papers also reported that providing group-housed ewes with lambing cubicles reduced incidences of lamb stealing [45, 46]. However, other papers reported that providing hides increased injuries due to competition [69], resulted in lower fox fur quality [64], decreased feed intake [65], and decreased growth [78]. Research is still needed to address these issues while still allowing the animals to express their natural hiding behaviours.

**Physiological indicators of stress.**   Several papers reported outcomes related to physiological indicators of stress, including 8 positive outcomes, 1 negative, and 2 neutral outcomes. For example, farrowing sows had decreased heart rate when housed in a pen that allowed for seclusion [43], as did mink when housed with a nest box [123] and fish when given hiding enrichment [126]. Chicken [127] and quail [118] hens had reduced plasma and fecal glucocorticoid levels and decreased tonic immobility when housed with nest boxes, while duck hens experienced "stress-induced hyperthermia" when unable to access their nest boxes [89]. However, two papers reported that providing hens with nest boxes did not have an impact on physiological indicators of stress, such as corticosterone levels [128, 129]. An additional study reported that providing fish with hiding places increased cortisol due to the accompanying increase in competition and aggression [69]. Development of methods to practically provide sufficient hiding spaces and reduce the accompanying agonism and stress may help resolve some of these issues.

**Behavioural indicators of affective state.**   Implications about animals' affective state based on behavioural measures were also reported in many studies. From these, 23 positive outcomes, 4 negative outcomes, and 8 neutral outcomes were reported. Most of these papers used poultry; for example, it was documented that providing hens with nests increased settled laying behaviours [130, 131], reduced behavioural signs of frustration (pacing and sham dustbathing) [132, 133], and increased comfort behaviours (preening) [134]. One paper

demonstrated the offspring of quail hens housed with hiding walls had lower emotional reactivity, even though the offspring never experienced the hides themselves [105]. Hiding places improved maternal care in both fox [50], mink [62], and sheep [46] dams, and foxes showed decreased fear of humans [135]. Rabbit dams given a more secluded nest box performed fewer nest-disturbing behaviours, and rabbits housed with a hiding shelter performed fewer abnormal behaviours (restlessness, excessive grooming, bar-gnawing, and timidity) [80]. Additionally, group-housed dairy cows who were provided with a hiding wall showed more settled behaviour when close to calving, including paying less attention to other cows and fewer position changes [136]. Group-housed ewes who were provided with lambing cubicles also demonstrated more settled pre-lambing behaviours, and ewes who eventually selected to lamb inside a cubicle spent less time traveling and investigating birth sites. However, two papers reported that housing foxes with hiding places increased fearfulness [64, 137], and group-housed pigs' resting behaviour was disturbed when housed with hiding walls [121]. Most studies that included these outcomes reported a positive effect of the hiding place, but more research is encouraged to understand the behavioural indicators of negative affective state, such as increased fearfulness.

## Limitations

The search criteria required that papers include "behaviour" or "welfare" in the title or abstract, thus, papers focusing on physiological or production measures may be underrepresented in this paper. Moreover, several papers in this review (37/151), particularly for fur carnivores and poultry, provided the animals with hiding places in combination with other enrichment items, thus it is not clear if the outcomes were a result of the hide alone [138]. These papers still provide valuable insight into the housing needs of farmed animals, especially since several papers have reported that when animals are given several enrichment items, the hiding enrichment item is among the most interacted with [67], and their use remains high over time [68]. However, it is a limitation of this review that the impact of the hiding spaces in these studies cannot be separated from the other enrichment items also provided.

## Conclusion

In this scoping review, we investigated the existing research into the provision of hiding spaces to farmed animals, including the types of animals represented in this literature, the context of hide provision, and the reported impact that hides had on the animals. Many animal types were represented, with the most being laying hens. Some animals were underrepresented in the literature and others were missing entirely, despite evidence that many of these animals perform hiding behaviors in their natural habitat. Common situations in which animals were provided hides included at parturition and oviposition, for general enrichment, for neonatal animals, or for injured/sick animals; the latter two categories were underrepresented and require further investigation. We also categorized the outcomes reported in the papers and assessed their implications for how hiding places impact animals. Although most outcomes indicate that providing farmed animals is beneficial for their welfare, health or productivity, some conflicts exist in the literature. In cases where outcomes indicate that hiding places may have a negative impact on the animals, the authors often reported competition among group-housed animals for the hiding places. We emphasize that while providing farmed animals with hiding places appears to be beneficial to their welfare in many circumstances, it should not be considered a replacement to more systematic housing concerns such as adequate space allowance.

## Supporting information

**S1 Checklist. Preferred Reporting Items for Systematic reviews and Meta-Analyses extension for Scoping Reviews (PRISMA-ScR) checklist.**
(DOCX)

## Acknowledgments

We would like to acknowledge Kim Mears for her assistance in developing the literature search.

## Author Contributions

**Conceptualization:** Hannah B. Spitzer, Rebecca K. Meagher, Kathryn L. Proudfoot.

**Data curation:** Hannah B. Spitzer.

**Formal analysis:** Hannah B. Spitzer.

**Funding acquisition:** Rebecca K. Meagher, Kathryn L. Proudfoot.

**Investigation:** Hannah B. Spitzer, Rebecca K. Meagher, Kathryn L. Proudfoot.

**Methodology:** Hannah B. Spitzer, Rebecca K. Meagher, Kathryn L. Proudfoot.

**Project administration:** Hannah B. Spitzer.

**Supervision:** Rebecca K. Meagher, Kathryn L. Proudfoot.

**Visualization:** Hannah B. Spitzer.

**Writing – original draft:** Hannah B. Spitzer.

**Writing – review & editing:** Rebecca K. Meagher, Kathryn L. Proudfoot.

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
