## [Decision Letter · Decision Letter 0]

28 Jul 2022

PONE-D-22-15757The impact of providing hiding spaces to farmed animals: A scoping reviewPLOS ONE

Dear Dr. Proudfoot,

Thank you for submitting your manuscript to PLOS ONE. After careful consideration, we feel that it has merit but does not fully meet PLOS ONE’s publication criteria as it currently stands. Therefore, we invite you to submit a revised version of the manuscript that addresses the points raised during the review process. Your manuscript has been reviewed by two experts in the field that, although found the study meritorious, have highlighted some areas that require further work before this manuscript can move forward. I concur with their view. Please address carefully all reviewers' comments when/if resubmitting your revised manuscript.

We look forward to receiving your revised manuscript.

Kind regards,

Angel Abuelo, DVM, MRes, MSc, PhD, DABVP (Dairy), DECBHM

Academic Editor

PLOS ONE

Journal Requirements:

We would like to acknowledge Kim Mears for her assistance in developing the literature search. Funding was provided by Sir James Dunn Animal Welfare Centre. 

 HS was provided a stipend provided by the University of Prince Edward Island's Sir James Dunn Animal Welfare Centre, https://awc.upei.ca/

KP is Director of the Sir James Dunn Animal Welfare Centre and played a role in the study design, data collection and analysis, decision to publish and preparation of the manuscript. 

Reviewers' comments:

Reviewer's Responses to Questions

**Comments to the Author**

1. Is the manuscript technically sound, and do the data support the conclusions?

Reviewer #1: Yes

Reviewer #2: Yes

2. Has the statistical analysis been performed appropriately and rigorously? 

Reviewer #1: Yes

Reviewer #2: Yes

3. Have the authors made all data underlying the findings in their manuscript fully available?

Reviewer #1: Yes

Reviewer #2: Yes

4. Is the manuscript presented in an intelligible fashion and written in standard English?

Reviewer #1: Yes

Reviewer #2: Yes

5. Review Comments to the Author

Reviewer #1: Overall comments: Your scoping review pulls together a growing pool of literature to identify consensus and needs across farmed species. It is a well-written manuscript that warrants publication after some minor editing.

Line Number: Comment/Question:

45 It may be useful to choose to use either ‘welfare’ or ‘well-being’ exclusively in your manuscript. Otherwise, it may be valuable to indicate to the reader that you are using the terms synonymously.

74 Here and throughout, use past tense to describe actions you took in conducting your review. For example, “To be included, a study must have had at least…”

112-113 Here and throughout, change actions that were completed in the past to past tense.

127 Were there specific factors that helped the authors to determine if the overall outcome was positive, negative, or inconclusive/neutral? In its current form, this statement is subjective and may be difficult to replicate.

217-218 Please provide more explanation of the following statement: “Wild lobsters also make great use of hiding places throughout their lives.”

235 Consider replacing ‘stage’ with ‘time’ to remain consistent with earlier text.

386 Omit ‘them’.

426 Instead of stress in general, is distress a more appropriate term?

478 If possible, it would be useful to include the number of studies that introduced other enrichment that may be confounding.

Reviewer #2: This field of research is a huge undertaking, and the paper does a good job of describing the current literature. The topic is of importance to the scientific community and the results are novel. In general, with so many species I think it may be easy to get lost in the weeds. Something I think could strengthen this paper is a description of the function of hiding behavior in the introduction (or elsewhere if there is a better fit). For example, cattle separate from the herd and use environmental coverage at calving in semi-natural settings, pigs build themselves a nest before farrowing, and mink/foxes/rabbits live in burrows or dens. And then why is providing hiding opportunities for farmed species important? This could also use additional set-up in the introduction. Specific comments are below.

The abstract describes the methodology well but in general is missing a set-up of why opportunities to hide is important for farm species. Similarly, the rationale for describing negative/neutral outcomes is unclear without first describing if hides benefit animals. Specific comments are below

15-16: It is unclear here why the link is being made or how hiding affects farmed animals

17-20: I’m curious about the use of the term “hides”. It isn’t defined clearly in the abstract. Could a better term be “hiding spaces” if “hide” isn’t defined right away?

28-31: What do the % values refer to?

32-33: Why highlight the inconclusive/negative outcomes from the review without the positive?

33-35: To include this summary consider first describing the species, age, and health status results found in the scoping review

45: Consumers or the public?

46-48: This sentence is somewhat unclear in the direction of causation. It seems like naturalness in production is a problem because opportunities to perform natural behaviors aren’t provided and may be motivated to do so

49-55: The topic of hiding in all farm species is cumbersome but I think it would be worthwhile to expand on the function of hiding behavior in natural or semi-natural settings and why this topic is important enough for a scoping review (i.e., some animals are highly motivated to hide during specific events)

56: Again, why is the research in this field of study increasing?

59-63: I would suggest defining the term “hide” in the introduction prior to stating the objectives. Not all hiding spaces are referred to as hides in the literature (e.g., nest boxes for hens)

65: In general, the methods section is a little bit murky. I would suggest making the definitions for screening/inclusion for clear cut. For example, potentially identify the questions each title/abstract was assessed for, then what criteria made it an inclusion/exclusion to full text criteria. You may consider using bullet points or a numbered list for clarity

97: How many results were found for each search term?

114: Was a full text screening performed after the title/abstract screening?

120: Suggest leaving the number of papers out because it isn’t presented until the results section

120-121: What does “according to the aims of the paper” mean? This could use additional detail

122: The reason for animal categorization needs to be defined. How/why you decided on specific categories should really be described. For example, why not categorize animals into “mammal, bird, fish, etc.” or “herbivore, omnivore, carnivore”? If there are hiding behaviors specific to each group you have identified, make sure its clear in the methodology. Also, crustacean should be reflected in Figure 2

Line 125: Was sample size included?

126-127: The terms “positive, negative, neutral” are somewhat subjective. Its unclear if the authors have defined the impacts on the animals or they are reporting what the authors of the original articles determined were “positive/negative/neutral”. A better way to report outcomes may be by creating a definition for these terms first before describing the categorization. E.g., an outcome was determined to be “positive” if it reduced disease, was preferred by animals, or reduced the performance of abnormal behaviors

131-140: Make sure all these criteria are outlined in the methods. This is the first time full text review and hand-searching papers is discussed

142: Why not report year of publication and country of study? These were included in the categorization (line 122)

182-184: What does this mean? If a study had multiple outcomes (e.g., behavior and production) each outcome was included in the table? Additionally, it may be helpful to describe the # of papers that had each of the 7 categorized outcomes

190/Table 2: Crustaceans should either be lobsters/crustaceans throughout to be consistent. What each of these categories mean is unclear at this point, I’m hoping the definitions/descriptions will be elaborated on in the discussion.

General comment on results: The results so far are interesting but feel as though they are lacking some meaning. I would suggest trying to dig in to the function of hiding behavior. Potentially this could be done by adding another table like table 2 with “context of hide” as the other category

207: Domestic animal behavior research?

211: 6-13 studies per species listed?

213: The 1 to 5 studies is confusing, like above, it may be clearer to provide the number of % of studies after each species

225: This is the first time these species are mentioned. They should be described in the methods and/or results

258: There are 2 published studies on the hiding behavior of ewes at parturition in “cubicles” - Use of Lambing Cubicles and the Behavior of Ewes at Parturition (Gonyou and Stookey, 1983) and Behavior of parturient ewes in group lambing pens with and without cubicles (Genyou and Stookey 1985)

332-335: The cattle described Lidfors (71) and Edwards papers (73) separated from other cattle and sought more ‘covered’ areas around parturition. However, neither of these groups of animals used man-made “hiding spaces” to calve. Here, and elsewhere as it applies, it may be more correct to describe the animal behavior rather than the defined environment

354: What kind of injury?

433: I’m curious about the injuries described here. Are these injuries from the environment, self-mutilation, aggressive interactions between animals, or something else?

450: Affective state needs to be defined somewhere

Figure 1: Does “Records screened” refer to the title/abstracts screened (same for records excluded)? Was a full text screening performed? And the final box, what is “reports of included studies”

Figure 2: Y-axis should be “number of studies per species”. All the species have a larger umbrella/category, would suggest giving lobsters and rabbit a category too to define why they don’t fit into any of the other categories.

6. PLOS authors have the option to publish the peer review history of their article (what does this mean?). If published, this will include your full peer review and any attached files.

Reviewer #1: **Yes: **Kurt D. Vogel

Reviewer #2: No

---

## [Author Response · Author response to Decision Letter 0]

15 Sep 2022

Reviewer 1: 

Overall comments: 

Your scoping review pulls together a growing pool of literature to identify consensus and needs across farmed species. It is a well-written manuscript that warrants publication after some minor editing.

Response: Thank you!

45 It may be useful to choose to use either ‘welfare’ or ‘well-being’ exclusively in your manuscript. Otherwise, it may be valuable to indicate to the reader that you are using the terms synonymously.

Response: We replaced the term ‘well-being’ with the term ‘welfare’ for improved clarity. 

74 Here and throughout, use past tense to describe actions you took in conducting your review. For example, “To be included, a study must have had at least…”

Response: Good catch! Present tense actions were replaced with past tense actions. 

112-113 Here and throughout, change actions that were completed in the past to past tense.

Response: Present tense actions were replaced with past tense actions. 

127 Were there specific factors that helped the authors to determine if the overall outcome was positive, negative, or inconclusive/neutral? In its current form, this statement is subjective and may be difficult to replicate.

Response: More details were added on the process of classifying outcomes as positive, negative, or inconclusive/neutral. 

217-218 Please provide more explanation of the following statement: “Wild lobsters also make great use of hiding places throughout their lives.”

Response: More details were added to this sentence. 

235 Consider replacing ‘stage’ with ‘time’ to remain consistent with earlier text.

Response: Good suggestion – ‘stage’ was replaced with ‘time’ to improve consistency throughout the text. 

386 Omit ‘them’.

Response: Good catch – ‘them’ was omitted. 

426 Instead of stress in general, is distress a more appropriate term?

Response: We changed lines 544-456 to read “It was commonly concluded that the reduction of abnormal repetitive behaviours was because hiding spaces helped reduce negative affective and physiological states that are presumed to be negative, such as perceived boredom, distress, or fear.”

478 If possible, it would be useful to include the number of studies that introduced other enrichment that may be confounding.

Response: The number of studies that provided hiding places in combination with other enrichment items was provided on line 568 (37 out of the 151 papers included in the review). 

Reviewer 2:

Overall comments: 

This field of research is a huge undertaking, and the paper does a good job of describing the current literature. The topic is of importance to the scientific community and the results are novel. In general, with so many species I think it may be easy to get lost in the weeds. Something I think could strengthen this paper is a description of the function of hiding behavior in the introduction (or elsewhere if there is a better fit). For example, cattle separate from the herd and use environmental coverage at calving in semi-natural settings, pigs build themselves a nest before farrowing, and mink/foxes/rabbits live in burrows or dens. And then why is providing hiding opportunities for farmed species important? This could also use additional set-up in the introduction. Specific comments are below.

Response: We thank the reviewer for their very thorough and thoughtful comments! We agree that the function of hiding behavior is a critical component of this review, so we have added background information on the natural hiding behaviour of wild animals on Lines 49-76. We have also added more information about why this is important for animal welfare on Lines 77-80. 

The abstract describes the methodology well but in general is missing a set-up of why opportunities to hide is important for farm species. Similarly, the rationale for describing negative/neutral outcomes is unclear without first describing if hides benefit animals. 

Response: We agree and have restructured the beginning of the abstract to now read: “Many wild animals will perform hiding behaviours for a variety of reasons, such as evading predators or other conspecifics. Unlike their wild counterparts, farmed animals are often housed in relatively barren environments without the opportunity to hide. Researchers have begun to study the impact of access to hides in farmed animals, including possible effects on animal welfare.” We are limited in word count for the abstract, so we hope that this concise set-up helps provide context for the review. We also did not want to assume that hides were good for animals until we completed the review (as this was the purpose of objective 3), so we have left this out and kept it more open. 

Specific comments are below. 

15-16: It is unclear here why the link is being made or how hiding affects farmed animals

Response: We have changed the wording of the abstract based on the reviewers earlier comment. 

17-20: I’m curious about the use of the term “hides”. It isn’t defined clearly in the abstract. Could a better term be “hiding spaces” if “hide” isn’t defined right away?

Response: Good point – the text has been modified to indicate that the terms “hiding space” and “hides” are being used synonymously. 

28-31: What do the % values refer to?

Response: Text has been added to clarify that the % values are indicating the % of papers included in the review. As we are limited on word count, we added this to the first % and left it off of the rest. 

32-33: Why highlight the inconclusive/negative outcomes from the review without the positive?

Response: Due to a limited word count we have opted to make this summary shorter so that Line 34-36 now reads: “Hides had a generally positive impact on animals, but more research is encouraged for under-represented species.” 

33-35: To include this summary consider first describing the species, age, and health status results found in the scoping review

Response: This sentence was again shortened to meet the word count – we have opted to leave this more open-ended and explain the gaps more in the discussion. Line 34-35 now reads: “Hides had a generally positive impact on the animals included in this review; more research is encouraged for under-represented species.” 

45: Consumers or the public?

Response: Good comment – we changed the term “consumer” to the “citizen” to better capture the meaning from the cited papers.

46-48: This sentence is somewhat unclear in the direction of causation. It seems like naturalness in production is a problem because opportunities to perform natural behaviors aren’t provided and may be motivated to do so

Response: The wording of this sentence was changed for improved clarity.

49-55: The topic of hiding in all farm species is cumbersome but I think it would be worthwhile to expand on the function of hiding behavior in natural or semi-natural settings and why this topic is important enough for a scoping review (i.e., some animals are highly motivated to hide during specific events)

Response: We agree with this comment and have added a paragraph describing the possible functions of hiding behaviors in species that are related to the farmed species included in this review (Lines 49-76).

56: Again, why is the research in this field of study increasing?

Response: Additional information was added to describe how allowing farmed animals to perform natural behaviours can benefit welfare. Lines 77-82 now reads: “Allowing farm animals the opportunity to perform natural behaviours, such as hiding, is an important component of animal welfare for many stakeholders. Moreover, allowing the performance of some natural behaviors may also improve other aspects of welfare, such as affective states and biological functioning (ref). Because of these potential benefits, there has been increasing research on allowing farmed animals to perform natural hiding behaviours, despite few conventional management systems that currently allow for this behaviour.”

59-63: I would suggest defining the term “hide” in the introduction prior to stating the objectives. Not all hiding spaces are referred to as hides in the literature (e.g., nest boxes for hens)

Response: Good point here, we added a description of what we considered a hide into this section. 

65: In general, the methods section is a little bit murky. I would suggest making the definitions for screening/inclusion for clear cut. For example, potentially identify the questions each title/abstract was assessed for, then what criteria made it an inclusion/exclusion to full text criteria. You may consider using bullet points or a numbered list for clarity

Response: We have changed this section so that our inclusion criteria is now a numbered list to help for clarity. 

97: How many results were found for each search term?

Response: We clarified in our methods section that the searches were ran as a single search for all search terms. Because of this, results for each individual search term are not available. 

114: Was a full text screening performed after the title/abstract screening?

Response: Yes – we added details about our full text screening process. 

120: Suggest leaving the number of papers out because it isn’t presented until the results section

Response: The total number of papers was removed from this line. 

120-121: What does “according to the aims of the paper” mean? This could use additional detail

Response: The wording of this sentence was altered to improve clarity so that lines 151-152 now read: “The 151 papers included in the final review were coded to facilitate analysis of the aims of this paper.” 

122: The reason for animal categorization needs to be defined. How/why you decided on specific categories should really be described. For example, why not categorize animals into “mammal, bird, fish, etc.” or “herbivore, omnivore, carnivore”? If there are hiding behaviors specific to each group you have identified, make sure its clear in the methodology. Also, crustacean should be reflected in Figure 2

Response: Details about how and why we selected these categories have been added into this section, so that Lines 156-162 now read: “We chose to use these specific “animal types” due to their relevance to farm animal industries. We also opted to further group them into “animal categories” based on biological and behavioural similarities; for example, we used the broader category of ungulate to represent cattle, deer, pigs, sheep, and goats because these animals have similar natural histories. However, rabbits and lobsters were the only animal types found in our search that fit in the small herbivore and crustacean categories.” The figures have also been changed to be aligned with the changes made in this section. 

Line 125: Was sample size included?

Response: Sample size may be relevant to include for a systematic review; however, we chose not to include sample size because this is a less detailed review.

126-127: The terms “positive, negative, neutral” are somewhat subjective. Its unclear if the authors have defined the impacts on the animals or they are reporting what the authors of the original articles determined were “positive/negative/neutral”. A better way to report outcomes may be by creating a definition for these terms first before describing the categorization. E.g., an outcome was determined to be “positive” if it reduced disease, was preferred by animals, or reduced the performance of abnormal behaviors

Response: We have added much more details on the process of classifying outcomes as positive, negative, or inconclusive/neutral. 

131-140: Make sure all these criteria are outlined in the methods. This is the first time full text review and hand-searching papers is discussed

Response: Details on the full text review and hand searching were added to the methods section. 

142: Why not report year of publication and country of study? These were included in the categorization (line 122)

Response: Good point - we added the range of publication years in the results section to demonstrate that research in this area is increasing over time. We removed country of study since, while interesting, we don’t feel it contributes to our aims. 

182-184: What does this mean? If a study had multiple outcomes (e.g., behavior and production) each outcome was included in the table? Additionally, it may be helpful to describe the # of papers that had each of the 7 categorized outcomes

Response: We have added language at the start of this section to describe how multiple outcomes were counted (lines 248-249). We also added a description of the number of papers that reported each outcome category (lines 260-266). 

190/Table 2: Crustaceans should either be lobsters/crustaceans throughout to be consistent. What each of these categories mean is unclear at this point, I’m hoping the definitions/descriptions will be elaborated on in the discussion.

Response: Further detail has been added into the paper to improve the meaning of our categories. The crustacean/lobster inconsistencies have been corrected. 

General comment on results: The results so far are interesting but feel as though they are lacking some meaning. I would suggest trying to dig in to the function of hiding behavior. Potentially this could be done by adding another table like table 2 with “context of hide” as the other category

Response: We like this idea and have added the context of hide provision into table 2 for each animal category. 

207: Domestic animal behavior research?

Response: Good suggestion, we modified the wording of this sentence for improved clarity. 

211: 6-13 studies per species listed?

Response: Yes, we added clarification to the text. 

213: The 1 to 5 studies is confusing, like above, it may be clearer to provide the number of % of studies after each species

Response: We modified the wording to include the number of studies after each animal type for improved clarity. 

225: This is the first time these species are mentioned. They should be described in the methods and/or results

Response: These species were mentioned in the results section on lines 221-223: “There were 6 animal types included in the initial literature search for which no results were found: bison, elk, alpacas, llamas, horses, and donkeys.”

258: There are 2 published studies on the hiding behavior of ewes at parturition in “cubicles” - Use of Lambing Cubicles and the Behavior of Ewes at Parturition (Gonyou and Stookey, 1983) and Behavior of parturient ewes in group lambing pens with and without cubicles (Genyou and Stookey 1985)

Response: Thank you for the excellent suggestion of additional relevant papers, we have added these papers into the review. 

332-335: The cattle described Lidfors (71) and Edwards papers (73) separated from other cattle and sought more ‘covered’ areas around parturition. However, neither of these groups of animals used man-made “hiding spaces” to calve. Here, and elsewhere as it applies, it may be more correct to describe the animal behavior rather than the defined environment

Response: We like this idea and have added clarification to what kinds of hiding spaces were used by these animals and the specific behaviors that were performed. 

354: What kind of injury?

Response: Text was added to clarify that this paper explored disbudding-related injuries. 

433: I’m curious about the injuries described here. Are these injuries from the environment, self-mutilation, aggressive interactions between animals, or something else?

Response: More detail about the types of injuries was added. 

450: Affective state needs to be defined somewhere

Response: Good suggestion, we added in a brief definition of affective state in the introduction. 

Figure 1: Does “Records screened” refer to the title/abstracts screened (same for records excluded)? Was a full text screening performed? And the final box, what is “reports of included studies”

Response: In Figure 1, records screened refers to the title and abstract screening, and reports assessed refers to the full text review. The process for the title/abstract screening and the full text review have been clarified in the methods. “Reports of included studies” should be the same number as the number as “studies included in the review”, since there were no papers found that reported the same study data. This has been corrected in the figure. 

Figure 2: Y-axis should be “number of studies per species”. All the species have a larger umbrella/category, would suggest giving lobsters and rabbit a category too to define why they don’t fit into any of the other categories.

Response: The axis and umbrella categories were corrected, and an explanation for the lack of greater categories for rabbits and lobsters was added into the methods.

---

## [Decision Letter · Decision Letter 1]

2 Oct 2022

PONE-D-22-15757R1The impact of providing hiding spaces to farmed animals: A scoping reviewPLOS ONE

Dear Dr. Proudfoot,

Thank you for submitting your manuscript to PLOS ONE. After careful consideration, we feel that it has merit but does not fully meet PLOS ONE’s publication criteria as it currently stands. Therefore, we invite you to submit a revised version of the manuscript that addresses the points raised during the review process. Please submit your revised manuscript by Nov 16 2022 11:59PM. If you will need more time than this to complete your revisions, please reply to this message or contact the journal office at plosone@plos.org. Please include the following items when submitting your revised manuscript:A rebuttal letter that responds to each point raised by the academic editor and reviewer(s). You should upload this letter as a separate file labeled 'Response to Reviewers'.A marked-up copy of your manuscript that highlights changes made to the original version. You should upload this as a separate file labeled 'Revised Manuscript with Track Changes'.An unmarked version of your revised paper without tracked changes. You should upload this as a separate file labeled 'Manuscript'.If applicable, we recommend that you deposit your laboratory protocols in protocols.io to enhance the reproducibility of your results. Protocols.io assigns your protocol its own identifier (DOI) so that it can be cited independently in the future. For instructions see: https://journals.plos.org/plosone/s/submission-guidelines#loc-laboratory-protocols. Additionally, PLOS ONE offers an option for publishing peer-reviewed Lab Protocol articles, which describe protocols hosted on protocols.io. Read more information on sharing protocols at https://plos.org/protocols?utm_medium=editorial-email&utm_source=authorletters&utm_campaign=protocols.

We look forward to receiving your revised manuscript.

Kind regards,

Angel Abuelo, DVM, MRes, MSc, PhD, DABVP (Dairy), DECBHM

Academic Editor

PLOS ONE

Journal Requirements:

Reviewers' comments:

Reviewer's Responses to Questions

**Comments to the Author**

1. If the authors have adequately addressed your comments raised in a previous round of review and you feel that this manuscript is now acceptable for publication, you may indicate that here to bypass the “Comments to the Author” section, enter your conflict of interest statement in the “Confidential to Editor” section, and submit your "Accept" recommendation.

Reviewer #2: (No Response)

2. Is the manuscript technically sound, and do the data support the conclusions?

Reviewer #2: Yes

3. Has the statistical analysis been performed appropriately and rigorously? 

Reviewer #2: N/A

4. Have the authors made all data underlying the findings in their manuscript fully available?

Reviewer #2: Yes

5. Is the manuscript presented in an intelligible fashion and written in standard English?

Reviewer #2: Yes

6. Review Comments to the Author

Reviewer #2: The paper has improved substantially. Thank you to the authors for such thorough revisions. There are only a few minor comments to address:

58 – 60: Does this mean mink and foxes also hide their offspring to avoid predation and infanticide?

164: In this paper, there are multiple references to “themes” or “themes that emerged”. Was there a specific thematic analysis applied to the papers to determine themes present in the reviewed papers? Or were themes determined by the greatest % of outcomes? You may consider explaining how “themes” were defined

226 – 228: Could the “these animals were represented” be changed to something more specific to the figure? Ideally, someone should be able to interpret a figure with only the figure description without needing to read the paper

240: Same comment as above regarding the figure description

290: The reference to chickens being one of the ‘most widely used food animals’ could benefit from some content/explanation. Are they common in Canada, North America, or globally? Additionally, is the reference to broilers or laying hens

295: There are 2 commas between “turkeys” and “and”

295 – 297: In terms of a lack of research, is it specific to animals in natural or farmed settings?

302: Should the period be after 40?

315-316: Should the number of parturition/oviposition papers be the same? I.e., 80/151 and 63/84

477-479: Missing a reference?

Many of the paragraphs in the discussion end with – more work is need (302-303, 310-312, 347-349, 406-408, 432-433, 455-456, 461-462, 471-472, 480-481, 490-491, 507-508, 527-529, 540-542, 562-564). It appears one of the main conclusions of the review is that many areas of study need additional research. However, I’m not sure it warrants stating so frequently and some of the individual paragraph summaries could be more insightful to the topic at hand.

580: Does chickens refer to broilers or laying hens?

7. PLOS authors have the option to publish the peer review history of their article (what does this mean?). If published, this will include your full peer review and any attached files.

Reviewer #2: No

---

## [Author Response · Author response to Decision Letter 1]

24 Oct 2022

The paper has improved substantially. Thank you to the authors for such thorough revisions. There are only a few minor comments to address:

58 – 60: Does this mean mink and foxes also hide their offspring to avoid predation and infanticide?

Answer: Yes, this was the intended meaning. We slightly modified the wording to improve clarity. 

164: In this paper, there are multiple references to “themes” or “themes that emerged”. Was there a specific thematic analysis applied to the papers to determine themes present in the reviewed papers? Or were themes determined by the greatest % of outcomes? You may consider explaining how “themes” were defined

Answer: We did not perform a thematic analysis, and we modified the wording of this sentence to better reflect our methodology.

226 – 228: Could the “these animals were represented” be changed to something more specific to the figure? Ideally, someone should be able to interpret a figure with only the figure description without needing to read the paper

Answer: We appreciate this suggestion, and we modified the wording of the figure to be more comprehensible as a stand-alone description of the figure.

240: Same comment as above regarding the figure description

Answer: We similarly modified the wording of this figure. 

290: The reference to chickens being one of the ‘most widely used food animals’ could benefit from some content/explanation. Are they common in Canada, North America, or globally? Additionally, is the reference to broilers or laying hens

Answer: Clarity was added into the sentence for both questions. 

295: There are 2 commas between “turkeys” and “and”

Answer: Thank you, this has been resolved. 

295 – 297: In terms of a lack of research, is it specific to animals in natural or farmed settings?

Answer: We were referring to lack of research in farmed settings, this has been clarified. 

302: Should the period be after 40?

Answer: The unneeded references after the period were removed, resolving the seemingly incorrect period placement. 

315-316: Should the number of parturition/oviposition papers be the same? I.e., 80/151 and 63/84

Answer: Thank you for catching this, they should be the same. The numbers 80/151 were corrected to 84/151. 

477-479: Missing a reference?

Answer: Good catch – the reference was added in. 

Many of the paragraphs in the discussion end with – more work is need (302-303, 310-312, 347-349, 406-408, 432-433, 455-456, 461-462, 471-472, 480-481, 490-491, 507-508, 527-529, 540-542, 562-564). It appears one of the main conclusions of the review is that many areas of study need additional research. However, I’m not sure it warrants stating so frequently and some of the individual paragraph summaries could be more insightful to the topic at hand.

Answer: We appreciate the identification of repetitive areas of the manuscript and have reviewed the highlighted portions. We are choosing to retain many of these sentences as identification of gaps in the literature was one of our main objectives; however, we modify or eliminate some of these phrases (lines 306-307, 494-495, and 544-546 in the manuscript with tracked changes). 

580: Does chickens refer to broilers or laying hens?

Answer: This was referring to laying hens, and this was added into the sentence for better clarity.

---

## [Editor Report · Decision Letter 2]

2 Nov 2022

The impact of providing hiding spaces to farmed animals: A scoping review

PONE-D-22-15757R2

Dear Dr. Proudfoot,

We’re pleased to inform you that your manuscript has been judged scientifically suitable for publication and will be formally accepted for publication once it meets all outstanding technical requirements.

Kind regards,

Angel Abuelo, DVM, MRes, MSc, PhD, DABVP (Dairy), DECBHM

Academic Editor

PLOS ONE
---

## [Editor Report · Acceptance letter]

16 Nov 2022

PONE-D-22-15757R2 

The impact of providing hiding spaces to farmed animals: A scoping review 

Dear Dr. Proudfoot:

I'm pleased to inform you that your manuscript has been deemed suitable for publication in PLOS ONE. Congratulations! Your manuscript is now with our production department. 

Kind regards, 

on behalf of

Dr. Angel Abuelo 

Academic Editor

PLOS ONE